# A Multiplex Serological Assay to Evaluate the Antibody Responses to a Set of *Plasmodium falciparum* Antigens and Their Protective Role Against Malaria in Children Aged 1.5 to 12 Years Living in a Highly Seasonal Malaria Transmission Area of Burkina Faso

**DOI:** 10.3390/vaccines13111091

**Published:** 2025-10-24

**Authors:** Sem Ezinmegnon, Issa Nébié, Tegwen Marlais, Daouda Ouattara, Amidou Diarra, Catriona Patterson, Kevin Tetteh, Alphonse Ouédraogo, Chris Drakeley, Alfred B. Tiono, Sodiomon B. Sirima

**Affiliations:** 1Groupe de Recherche Action en Santé (GRAS), Ouagadougou 06 BP 10248, Burkina Faso; i.ouedraogo@gras.bf (I.N.); d.ouattara@gras.bf (D.O.); a.diarra@gras.bf (A.D.); a.ouedraogo@gras.bf (A.O.); a.tiono@gras.bf (A.B.T.); s.sirima@gras.bf (S.B.S.); 2London School of Hygiene & Tropical Medicine, London WC1E 7HT, UK; tegwen.marlais@lshtm.ac.uk (T.M.); catriona.patterson@lshtm.ac.uk (C.P.); chris.drakeley@lshtm.ac.uk (C.D.); 3Foundation for Innovative New Diagnostics (FIND), Global Health Campus, 1202 Geneva, Switzerland; kevin.tetteh@finddx.org

**Keywords:** children, malaria, protective immunity, multiplex, IgG responses

## Abstract

**Background/Objectives:** Understanding the seroepidemiology of *P. falciparum* antibody responses is essential for assessing the acquisition of natural immunity and may guide interventions that impact the acquisition of immunity against malaria in endemic areas. This study assessed the association between antigen-specific IgG responses and protection against *P. falciparum* infection in children from Burkina Faso. **Methods**: Children aged 1.5 to 12 years were followed using cross-sectional and longitudinal approaches. IgG responses to 16 *P. falciparum* antigens were measured using a multiplex assay and analyzed by age group and malaria infection status. Associations between antibody levels and clinical malaria risk were assessed using incidence rate ratios (IRRs), and predictive performance of antibody combinations was evaluated using ROC analysis. **Results**: IgG responses to AMA1, CSP, and MSP2 CH150 showed weak but significant positive correlation with age. Children aged 5–12 years had higher antibody levels than younger children aged 1.5–5 years. Uninfected children had higher levels of antibodies to EBA181 RIII-V, Rh5.1, and SEA1, while infected children had elevated AMA1 and MSP2 CH150. Anti-GLURP R2 and anti-Rh5.1 antibodies were associated with reduced malaria risk (adjusted IRR = 0.52 and 0.40, respectively). The antibody combination of AMA1, GLURP R2, and Etramp5 Ag1 showed the best predictive performance (AUC = 0.70). **Conclusions**: This study underlines the value of less-studied antibodies (Etramp5 Ag1, Rh5.1, HSP40 Ag1) for diagnosing and protecting against malaria, opening up prospects for the development of more effective tests and targeted vaccine approaches. The variability of responses according to age and infection status calls for further studies to optimize prevention strategies.

## 1. Introduction

Despite the massive deployment of malaria control measures (malaria chemoprevention, indoor residual sprays and long-lasting insecticidal nets) in these last two decades, malaria remains a public health problem, mainly in sub-Saharan Africa, which bears 94% of the global malaria burden. In the 2023 World Malaria Report, the World Health Organization (WHO) estimated that 263 million cases of malaria occurred in 83 malaria-endemic countries, with a notably high mortality and morbidity rate among children under 5 years of age and pregnant women [1]. According to WHO, Burkina Faso was among the 10 high malaria burden countries in 2023, accounting for 3.1% of malaria cases worldwide and 2.7% of malaria deaths in 2023 [1].

In malaria-endemic regions, particularly within Africa, where *P. falciparum* is widespread, people are frequently exposed to this parasite, and most infected adults rarely develop severe malaria. There is evidence that antibodies, particularly immunoglobulin G (IgG) targeting various antigens at different developmental stages of *P. falciparum*, are crucial for developing immunity to malaria [2,3]. This immunity typically builds gradually through repeated exposures [4,5]. Although the association between exposure and protection against severe forms of malaria in adults is widely established, our understanding of how immunity to malaria is acquired remains limited, particularly in young children living in endemic areas and exposed from a very young age. Numerous studies have identified several *P. falciparum* antigens involved in the pathogenesis of malaria, and some of them are promising candidates for vaccines aimed at specifically targeting the immune response in individuals living in malaria-endemic areas. Key proteins such as CSP (Circumsporozoite Protein), AMA1 (Apical Membrane Antigen 1), MSP2 (Merozoite Surface Protein 2), EBA175 (Erythrocyte binding antigen), GLURP (Glutamate Rich Protein), and Rh5 (Reticulocyte Binding Protein Homolog 5) have been extensively investigated, with clinical studies demonstrating their efficacy as vaccine candidates [6,7,8,9,10,11,12]. Notably, the RTS,S and R21 vaccines, which utilize CSP antigens, have received approval from the World Health Organization (WHO) for deployment in children living in malaria-endemic areas starting in 2023 [13].

In addition to these well-characterized antigens, several lesser-studied *P. falciparum* proteins, such as Etramp (Early Transcribed Membrane Proteins), HSP40 Ag1 (Heat Shock Protein 40 Antigen 1), HYP2 (Hypothetical Protein 2), GEXP18 (Gametocyte-Exported Protein 18), SEA1 (Sporozoite Exported Antigen 1), and SBP1 (Sporozoite Binding Protein 1), have been increasingly incorporated into recent seroepidemiological studies. However, these antigens have mainly been characterized as markers of recent exposure to infection rather than protective targets, as there is currently no evidence supporting their efficacy as vaccine candidates. These antigens are expressed during the blood stage of the *P. falciparum* life cycle and may hold potential for future vaccine development [14,15,16]. Their relevance extends beyond vaccine formulation; they provide valuable insights into malaria immunity and transmission dynamics [17,18,19,20]. The correlation between these antigens and infection status underscores their importance as recent exposure markers in evaluating public health interventions aimed at controlling malaria. While protective measures such as seasonal malaria chemoprevention (SMC) and vaccines like RTS,S and R21 are being implemented, analyzing the seroepidemiology of anti-malaria antibodies is essential for enhancing intervention programs. Although specific antibodies against parasite antigens vary significantly in their longevity, monitoring antibody status serves as a sensitive indicator of immunity levels and allows for retrospective assessments of exposure history, including the impact of interventions and the absence of recent exposure in malaria elimination contexts.

Nested in a study originally designed to assess the malaria morbidity in children living in future malaria vaccine candidate trial sites, this sub-study aims to measure antibody responses against a panel of 16 *P. falciparum* antigens that include pre-erythrocytic, asexual blood, and sexual blood stage antigens and their association with protection against malaria infection in children during high malaria transmission season. The focus will particularly be on lesser-known antigens that have been less studied in seroprevalence research, with the goal of understanding their association with protection against malaria infection.

## 2. Materials and Methods

### 2.1. Study Site, Population, Design, and Period

The study was conducted in an area covered by the Banfora Health District, located in the southwestern part of Burkina Faso. The climate is characterized by a rainy season from May to November and a dry season from December to April. Malaria transmission is stable throughout the year but peaks during the rainy season (June to October). The Banfora health district has intense seasonal malaria transmission over the six-month period following seasonal rains from May to November [21]. *P. falciparum* accounts for more than 90% of malaria cases [22]. Malaria control and prevention interventions in the study area include SMC, which has been implemented since 2014 [23], along with the use of long-lasting insecticide-treated nets (LLINs). SMC targets children aged 3 to 59 months and consistently achieves high coverage in the health district [23,24].

The primary study was designed to assess the incidence of clinical malaria among children aged 1.5 to 12 years residing in the study area during the high malaria transmission season, and key findings were published (methodology published, PMID: 39695728). To ensure comparability, all participants were enrolled at the same time point during the peak of the transmission season (September 2020) and followed up prospectively using standardized active and passive surveillance protocols up to April 2021 (malaria low transmission period). Prior to follow-up, all participants were treated and confirmed negative for *P. falciparum* by PCR. Finger-prick samples for *P. falciparum* detection (microscopy and PCR) and venous blood samples for immunological analysis were collected. Afterwards, participants were followed up using active and passive case detection methods. Active methods consisted of biweekly home visits during which, after clinical evaluations, finger-prick blood samples were collected to detect incident infections. For passive surveillance, study participants attending local healthcare facilities were assessed by study staff for clinical malaria episode detection. A clinical malaria episode was defined by fever (axillary temperature ≥ 37.5 °C/tympanic temperature ≥ 38 °C or a history of fever in the past 24 h) associated with a positive thick smear/blood film for *P. falciparum*.

### 2.2. Sample Collection

Individual finger-prick blood samples were collected to perform blood smears on slides and dry blood spots (DBS) on filter paper (Whatman 3 mm, GE Healthcare, Pittsburg, PA, USA) from all children at enrollment at the peak of malaria transmission. DBS were labeled, then air-dried and placed individually in a plastic bag containing a desiccant to protect them from humidity, and were used for molecular and serological analysis. The blood smears were air-dried, the thin film fixed with methanol, and both smears stained with Giemsa (Merck, Darmstadt, Germany).

### 2.3. Plasmodium falciparum Infection Diagnosis

We performed *P. falciparum* infection diagnosis by both microscopy and PCR. For microscopy diagnosis, Giemsa-stained blood films were examined under 100× oil immersion magnification. Each slide was read twice independently by two different competent microscopists according to GRAS internal SOP. Malaria species and stages of development were assessed, and the parasitaemia was estimated in the thick blood films using an average count of 8000 white blood cells per microliter of blood. A slide was declared negative when no malaria parasite was seen after 200-higher power fields were examined.

From PCR, malaria infection, DNA was extracted from a quartered blood spot (17.5 µL) using the QIAamp Blood Mini Kit (QIAGEN, Germantown, MD, USA) according to the manufacturer’s instructions. The presence of *P. falciparum* DNA was assessed using a nested PCR protocol (nPCR) as described previously [25].

### 2.4. Multiplex Luminex Assay for Antibody Quantification

#### Antigen Panel

Antibody responses to a panel of antigens were measured (Appendix A). These included 16 antigens from sporozoite, merozoite, infected erythrocyte, and gametocyte stages of the *P. falciparum* parasite in the human host. Antigens were previously optimized for coupling conditions and covalently bound to magnetic microbeads (MagPlex, Luminex Corp., Austin, TX, USA) by the EDC/Sulfo-NHS intermediate reaction as described previously [18].

### 2.5. Antibody Detection Immunoassay

IgG antibody levels were determined as described elsewhere [18]. DBS were eluted in 200 µL of elution buffer B (1xPBS, 0.05% Tween, 0.5% BSA, 0.02% sodium azide, 0.1% casein, 0.5% polyvinyl alcohol (PVA), 0.5% polyvinyl pyrrolidone (PVP), 15.25 µg/mL *E. coli* lysate) overnight. For the assay procedure, an initial mixture containing 8 µL of each set of antigen-coupled microspheres and 5 mL of buffer A (PBS, 1% BSA, 0.02% NaN_3_, and 0.05% Tween-20) was prepared, yielding approximately 1000 beads per region per well. An amount of 50 µL of the bead mixture was added to a pre-wetted 96-well flat-bottom plate (BioPlex Pro™, Bio-Rad Laboratories, Watford, UK) and washed once by placing the assay plate onto a magnetic plate separator (Bio-Plex^®^, Bio-Rad Laboratories, Watford, UK) and pausing for 1 min. Plates were then inverted forcefully to remove the liquid, and 100 µL of PBS-T (1xPBS, 0.05% Tween-20) (Merck KGaA, Darmstadt, Germany) was added to each well. The wash was removed using a magnetic separator, and 50 µL of samples and controls were added to the plate and incubated in the dark at room temperature (RT) on a microplate shaker at 500 rpm for 90 min. A serially diluted hyperimmune pool (CP3) was included on each plate to assess inter-plate variability, malaria-naïve individuals to serve as negative controls, and two wells containing buffer diluent to serve as blanks. Following three washes, 50 µL of fluorescent secondary antibody (Jackson Immuno 109-116-098 (West Grove, Chester County, PA, USA): goat anti-human Fcy-fragment specific IgG conjugated to R-Phycoerythrin (R-PE)), diluted to a 1:200 dilution with buffer A, was added to all wells and incubated again for 90 min in the dark at RT at 500 rpm. After a further three washes, the plate was incubated in 50 µL of buffer A for 30 min. Plates received an additional wash and, after a final addition of 100 µL 1xPBS, were read using the Luminex MAGPIX^©^ bioanalyser (Luminex Corporation, Austin, TX, USA). At least 50 beads per analyte were acquired per sample, and the background-adjusted median fluorescence intensity (MFI) data were used for analysis.

### 2.6. Statistical Analysis

Statistical analyses were performed using R software version 3.6.1. Quantitative continuous IgG antibody response data (reported as log10 MFI values) were compared for different groups (i.e., by age group or malaria infection) using Student’s *t*-test or the Wilcoxon–Mann–Whitney test. Correlations between the levels of antibody responses and age were analyzed using Spearman’s rank correlation. For each antigen, negative binomial regression was used to investigate the association between the levels of antibody measured at baseline and the incidence rate of clinical malaria. The clinical malaria incidence rates were calculated as the number of episodes divided by the time at risk. A clinical malaria episode was defined based on different parasitaemia thresholds: the primary case definition was axillary temperature ≥ 37.5 °C/tympanic temperature ≥ 38 °C or a history of fever in the past 24 h associated with an asexual *P. falciparum* parasitaemia > 0. The secondary case definition was axillary temperature ≥ 37.5 °C/tympanic temperature ≥ 38 °C or a history of fever in the past 24 h associated with parasitaemia ≥ 5000 asexual *P. falciparum* stages parasites/µL. Children were monitored for 28 days after treatment to confirm both clinical recovery and parasitological clearance. To avoid double counting, any episode occurring within 28 days of a previous episode was excluded from the incidence calculation, and this 28 day window was also removed from the time-at-risk denominator. To account for potential confounding by recent exposure, infection status at the time of antibody measurement was included as a covariate in the adjusted regression models.

For predictive modeling of antibody combinations for protection against malaria infection, we systematically generated all possible antibody combinations ranging from simple combinations (pairs) to complex combinations (all antibodies) using the “combn” function in the “combinat” R package within RStudio (version 2025.09.1+401; Posit Software, PBC, Boston, MA, USA). For each combination, performance was assessed by analyzing three metrics: AUC (overall performance), sensitivity (detection of infected people), and specificity (avoidance of false positives). A weighted composite score (50% AUC, 30% sensitivity, 20% specificity) was used to objectively rank the combinations. The data is divided into training sets (70%) and test sets (30%) [26]. Logistic regression models were trained for each combination, validated by cross-validation (five folds), and assessed via AUC-ROC. Confidence intervals and DeLong tests guarantee robustness. The best combinations are selected to maximize the composite score, offering an optimal compromise for diagnosis or vaccine research.

## 3. Results

### 3.1. General Characteristics of the Study Population

A total of 474 participants were included in this study, with a similar gender distribution. The majority of children were over 5 years old (60.33%). At the beginning of the follow-up, 23% of children were infected with *P. falciparum* (microscopy), and 31% were found to be infected by *P. falciparum* (PCR), with prevalence increasing significantly with age (*p* < 0.001). The geometric mean density of *P. falciparum* asexual parasites was 743.35 parasites/µL (95% CI: 492.47–1122.03) of blood, which varied significantly between age groups (*p* < 0.001). Further details on the study population are shown in Table 1.

### 3.2. IgG Antibody Responses to Malaria Antigens and Age

IgG antibody responses against the malaria antigens AMA1, CSP, and MSP2 CH150 showed a statistically significant, although weak, positive correlation with age (r = 0.16, *p* < 0.001; r = 0.13, *p* = 0.01; and r = 0.22, *p* < 0.001, respectively) (Figure 1). Complementing these findings, a comparison within age groups revealed that children aged 5 to 12 years had significantly higher antibody levels against AMA1, CSP, and MSP2 CH150 than those aged 1.5 to 5 years (*p* < 0.05), reinforcing the trend of increasing antibody levels with age for these specific antigens.

When analyzing the data according to active *P. falciparum* infection by microscopy (parasitaemia > 0 asexual forms/µL of blood), antibody levels against EBA181 RIII-V, GEXP18, GLURP R2, HSP40 Ag1, HYP2, Rh5.1, SBP1, and SEA1 were significantly higher in uninfected children compared to infected children (*p* < 0.05). Conversely, antibody levels against AMA1 and MSP2 CH150 were significantly higher in infected children compared to uninfected children (*p* < 0.05). Antibodies against CSP, EBA175 RIII-V, and MSP2 D2 showed no significant differences between the two groups whatever the method of diagnosis (Figure 2).

### 3.3. Antibody Levels and Risk of P. falciparum Malaria Episodes

We evaluated the association between specific antibody levels at baseline and the risk of subsequent clinical malaria episodes, considering different case definitions described above. In total, 77 episodes and 53 episodes meeting, respectively, the secondary and the primary cases definitions were analyzed. The incidence rate ratios (IRR), both crude and age-adjusted, along with their 95% confidence intervals (CI), are presented in Table 2. Anti-AMA1 antibodies showed a marginally significant association with increased malaria risk: for the secondary case definition, the adjusted IRR was 1.28 (95% CI: 0.97–1.71; *p* = 0.06), with a similar trend for the primary case definition (IRR = 1.36; *p* = 0.06). In contrast, antibodies against GLURP R2 were significantly associated with protection for the secondary case definition (adjusted IRRs of 0.52; *p* = 0.04). Antibodies against Rh5-1 also showed a protective trend (adjusted IRR = 0.40), although this association was marginally significant (*p* = 0.07). Similarly, SEA1 antibodies suggested a potential protective effect (IRR = 0.40; *p* = 0.05). However, this protective effect disappeared for the primary case definition. No clear association was observed for other major antigens such as CSP, EBA175, or MSP2 variants. Some less well-studied antibodies, such as Etramp4 Ag2 (IRR = 0.53; *p* = 0.008) and HSP40 Ag1 (IRR = 0.49; *p* = 0.07), indicated a potential protective trend but did not reach statistical significance.

### 3.4. Predictive Modeling of Antibody Responses for Clinical Protection Against Malaria

For antibodies whose increased levels were associated with a reduced risk of malaria episodes, we used a machine learning methodology to identify the most promising antibody combinations for predicting protection against clinical malaria. Figure 3 presents the performance of the top 20 antibody combinations. The best-performing combinations (AUC > 0.68) systematically include Etramp5 Ag1, one of the antigens that has been little studied. Evaluation by composite score (weighting: 50% AUC, 30% sensitivity, 20% specificity) shows that the AMA1 + GLURP R2 + Etramp5 Ag1 antibody combination represents the best compromise between performance and complexity, with an optimal composite score of 0.70 and an AUC of 0.70 (IC95%: 0.6379–0.8514). In particular, the inclusion of Etramp5 Ag1 significantly increased sensitivity by 15 percentage points (81% vs. 66% for AMA1 + GLURP R2 alone). The AMA1 + GLURP R2 + HSP40 Ag1 + Etramp5 Ag1 combination also showed a performance/specificity compromise (AUC 0.69, composite score 0.69). The addition of HSP40 Ag1 to the AMA1 + GLURP R2 + Etramp5 Ag1 combination improves the AUC by 2.7% (0.678 to 0.690).

## 4. Discussion

This study provides valuable insights into the dynamics of antibody responses to a set of *P. falciparum* antigens in children from a malaria-endemic region of Burkina Faso. Among 474 children, antibody levels to certain antigens, particularly AMA1, CSP, and MSP2 CH150, showed modest but significant increases with age, suggesting gradual immune acquisition through repeated exposure. Interestingly, *P. falciparum* infection also affects antibody profiles, with some antigens elevated in uninfected children, while others, like AMA1, were higher in infected children. Using predictive modeling, combinations of antibodies, especially those including Etramp5 Ag1, GLURP R2, and AMA1, emerged as promising markers of clinical protection, highlighting the potential of multiplex serological signatures for malaria risk stratification.

We observe that antibodies against GLURP R2 were significantly associated with protection against clinical malaria, while SEA1 and Rh5.1 showed promising protective trends approaching statistical significance. Although antibodies against less-investigated antigens such as Etramp4 Ag2 and HSP40 Ag1 showed protective trends (IRR < 1), these associations did not reach statistical significance and require validation in larger studies. Furthermore, the combination of AMA1 + GLURP R2 + Etramp5 Ag1 emerged as the most predictive of protection, with a good balance between sensitivity and specificity, suggesting a new alternative in the design of multivalent malaria vaccines.

Although the *p*-values for the correlations with AMA1, CSP, and MSP2 CH150 were statistically significant, the weak correlation coefficients (*r* = 0.18, *r* = 0.13, and *r* = 0.21, respectively) suggest that the strength of the association between antibody responses and age may be limited. This reflects a gradual increase in immune responses with age, likely due to cumulative exposure to *P. falciparum* [27], but the overall relationship remains relatively modest. The same findings have been reported previously in other studies, showing that repeated exposure favors the development of a more robust antimalarial immunity response in older age groups and that antibodies can serve as markers of cumulative exposure where malaria is endemic [28,29,30]. However, this immunological maturation is restricted to certain antigens, since for 12 others, including several new targets analyzed, no significant association with age was detected. This indicates that the antibody responses observed for these antigens may reflect residual levels rather than age-specific immune maturation (Appendix A).

Studies of children who have active *P. falciparum* infection showed significantly higher levels of antibodies against the apical membrane antigen 1 (AMA1) and CH150/9 allele of MSP2 compared to those uninfected. AMA1 and the MSP family, including MSP2 antigens, are involved in the erythrocyte invasion and have been widely described as highly immunogenic antigens [30,31]. However, our finding suggests a paradoxical association: higher AMA1 antibody levels were linked to an increased risk of subsequent malaria episodes, as indicated by both crude and age-adjusted incidence rate ratio (IRR = 1.28–1.36). This observation may reflect the role of AMA1 as a marker of recent or repeated exposure rather than protective immunity [18,31].

On the other hand, the results show high titers of antibodies to specific *P. falciparum* antigens GLURP R2, MSP2 CH150, Rh5.1, and SBP1 in uninfected children compared with infected children, suggesting that these antibodies could play a protective role, potentially preventing infection or controlling the parasite load to an undetectable level. These well-characterized antigens are promising targets for vaccine development because of their involvement in the immune response against malaria [32,33,34]. The presence of antibodies against these less-studied antigens EBA181 RIII_V, GEXP18, HSP40 Ag1, and HYP2 in uninfected children opens up new possibilities for research. For example, a recent study showed that modifications to HSP40 Ag1 could affect its function and localization and play a role in the parasite’s stress response [35], while HYP2 and GEXP18 could be involved in virulence or immune escape mechanisms [18,36].

Analysis of incidence rate ratios (IRRs) revealed that antibodies targeting GLURP R2, Rh5.1, and SEA1 conferred significant protection at low parasitemia (adjusted IRR 0.40–0.52). This aligns with their established roles: Rh5.1 inhibits erythrocyte invasion [37], while GLURP R2 acts on early blood-stage parasites by blocking trophozoite-to-schizont differentiation [38]. For SEA1 (Schizont Egress Antigen-1), its association with protection at low parasite density suggests a role in inhibiting merozoite egress [19], though the precise mechanism remains to be elucidated. The loss of efficacy observed for these antibodies at high parasitemia may reflect antigen–antibody saturation or the emergence of parasite variants evading immune recognition [37,38]. These findings underscore the importance of accounting for parasitemia levels when evaluating antibody-mediated immunity to identify robust and versatile vaccine targets.

Our analysis demonstrated that a multi-antigen–antibody signature combining AMA1, GLURP R2, and Etramp5 Ag1 showed strong predictive value for malaria protection (AUC = 0.70, 95% CI: 0.59–0.81). This highlights the promise of multiparametric approaches in malaria immunology. Each component contributed a distinct protective mechanism: GLURP R2 mediated inhibition of parasite growth [38,39], while the less-studied Etramp5_ag1 served as a marker of submicroscopic infections [40]. Notably, adding AMA1, a known exposure marker, enhanced the sensitivity of the binary model (GLURP R2 + Etramp5 Ag1) by 14% (sensitivity increase from 0.66 to 0.81). This synergy suggests that simultaneously targeting early (GLURP) and late (Etramp5) blood-stage antigens, alongside an exposure marker (AMA1), provides broader intervention potential compared to single-antigen approaches.

The findings also highlight the emerging importance of antibodies targeting previously undervalued antigens, such as HSP40 Ag1 and Rh5.1, in enhancing malaria diagnostic performance. Studies conducted in low-transmission countries like Gambia, have demonstrated that antibodies against HSP40 Ag1 are significantly associated with clinical and asymptomatic *P. falciparum* infections [17], making them valuable serological markers for epidemiological surveillance. Similarly, antibodies against Rh5.1 have shown promising growth inhibitory activity in vitro, suggesting their potential not only in diagnostics but also in vaccine development [41]. The inclusion of these antibodies in combinations makes it possible to optimize both sensitivity and specificity, opening up new prospects for diagnosis and epidemiological surveillance markers in malaria elimination settings where the occurrence of submicroscopic infections might be very common. These findings highlight the importance of exploring combinations of antigens that act on complementary pathways within the same lifecycle stage, rather than focusing solely on individual targets, for vaccine development or the identification of protective biomarkers.

One of the paradoxical observations of our study is that AMA1, although a promising vaccine target [11,27,42], here shows an association with an increased risk of infection (IRR = 1.27–1.36). This observation could be explained by a few factors. First, the high immunogenicity of AMA1 primarily generates non-neutralizing antibodies, diverting the immune response away from protective epitopes [43,44]. Second, the allelic diversity of circulating AMA1 variants allows strains to evade recognition by existing antibodies [45]. Finally, the elevated antibody levels observed in infected children might reflect recurrent infections rather than protective immunity. These findings highlight the complexity of vaccine development and the need to better understand the underlying immunological mechanisms.

## 5. Conclusions

This study highlighted age-related differences and identified markers associated with protection against clinical malaria. Older children exhibit higher antibody levels, reflecting cumulative exposure and immune maturation. We observed that antibodies targeting some less-studied antibodies, such as HSp40 Ag1 and Etramp4_Ag2, showed trends suggestive of reduced malaria risk, although these associations did not reach statistical significance and warrant further investigation. Furthermore, machine learning analysis revealed a specific antibody combination and showed improved predictive performance for protection, suggesting potential value for the development of multivalent vaccines.

However, the study has certain limitations, including its focus on a single geographic area, the potential bias introduced by seasonal malaria chemoprevention (SMC) in children under five, which may have affected infection prevalence and antibody response estimates, and the lack of in-depth analysis of confounding factors such as genetic helminth infections and soil-transmitted helminths. These findings underscore the need for strategic approaches to vaccine development and call for further research to validate these observations in other epidemiological contexts and to explore deeply the mechanisms underlying this protective immunity.

## Figures and Tables

**Figure 1 vaccines-13-01091-f001:**
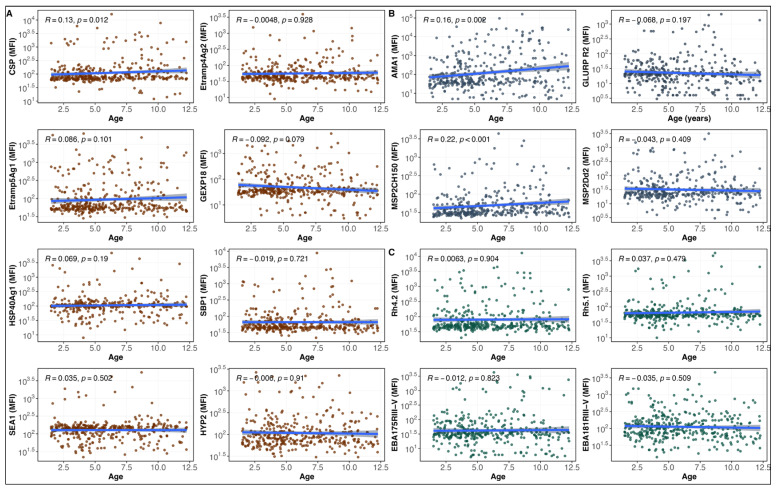
Baseline IgG levels in relation to the age of participants. Correlations between specific antibody responses to *P. falciparum* antigens and age, presented according to functional groupings: (**A**) antigens considered as markers of recent infection (CSP, Etramp4 Ag2, Etramp5 Ag1, GEXP18, HSP40 Ag1, SBP1, SEA1, HYP2), (**B**) markers of cumulative exposure (AMA1, GLURP R2, MSP2 CH150, MSP2 Dd2), and (**C**) proteins involved or thought to be involved in protective immunity (RH4.2, Rh5.1, EBA175 RIII-V, EBA181 RIII-V). Each graph shows the individual distribution of antibody response measurements (data points) according to the age, with a fitted linear regression line (solid blue line) and its 95% confidence interval (shaded area). Different colors represent different antigen groups: Orange/Brown for panel (**A**), blue/purple for panel (**B**), and teal/green for panel (**C**). Spearman’s correlation coefficients (R) and associated *p*-values are given for each antigen analyzed. MFI, Mean Fluorescence Intensity.

**Figure 2 vaccines-13-01091-f002:**
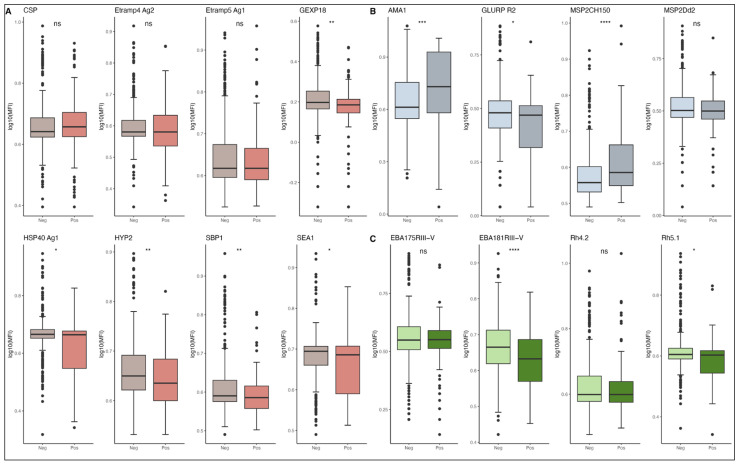
Baseline IgG levels in relation to active *P. falciparum* infection, presented according to *P. falciparum* antigens functional groupings: (**A**) antigens considered as markers of recent infection (CSP, Etramp4 Ag2, Etramp5 Ag1, GEXP18, HSP40 Ag1, SBP1, SEA1, HYP2), (**B**) markers of cumulative exposure (AMA1, GLURP R2, MSP2 CH150, MSP2 Dd2), and (**C**) proteins involved or thought to be involved in protective immunity (RH4.2, Rh5.1, EBA175 RIII-V, EBA181 RIII-V). Antibody MFI data were log_10_-transformed. Pos, microscopic malaria-positive individuals; Neg, uninfected individuals. Each box plot shows a distribution of antibody levels of each group, with box hinges indicate the 25th and 75th percentiles, the horizontal line showing the median, and whiskers extending to hinges ± 1.5 × IQR. Individual data points are overlaid as black dots. Different colors represent different antigen groups: Orange/Brown for panel (**A**), blue/purple for panel (**B**), and teal/green for panel (**C**), with lighter shades for uninfected and darker shades for infected individuals. *p*-values from unpaired Wilcoxon rank-sum tests are indicated above each comparison: ns, not significant; * *p* < 0.05; ** *p* < 0.01; *** *p* < 0.001; **** *p* < 0.0001. MFI, Mean Fluorescence Intensity; IQR, interquartile range.

**Figure 3 vaccines-13-01091-f003:**
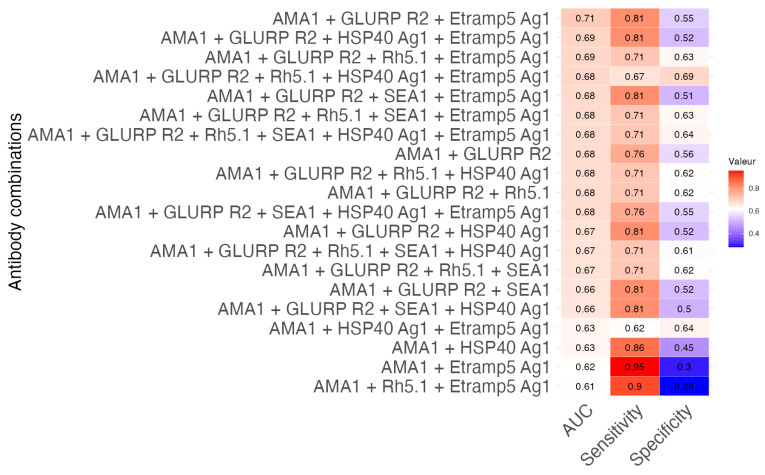
Predictive models using combined antibody response. The heat map displays Area Under the Curve (AUC) values for different combinations of antibody markers in predictive models. Colors represent AUC performance: blue indicates lower predictive accuracy, while red indicates higher predictive accuracy. Each cell shows the AUC value for the corresponding antibody combination, with sensitivity and specificity values provided in the right palette.

**Table 1 vaccines-13-01091-t001:** Characteristics of the study population by age groups.

Parameters	Total	1.5–5 Years	5–12 Years	*p*-Value ^2^
	*N* = 474 ^1^	*N* = 188 ^1^	*N* = 286 ^1^	
Gender (*n*, %)				0.8
M	233 (49%)	94 (50%)	139 (49%)	
F	241 (51%)	94 (50%)	147 (51%)	
Age (mean, range)	6.42 (1.51–12.27)	3.40 (1.51–4.98)	8.40 (5.06–12.27)	<0.001
Baseline asexual *Pf* infection (microscopy), (*n*, %)	109 (23%)	15 (8.0%)	94 (33%)	<0.001
Baseline asexual *Pf* parasite density (/µL of blood) *	743.35 (492.47–1122.03)	1350.34 (496.6–3675.79)	675.81 (431.41–1058.66)	<0.001
*Pf* infection by PCR (*n*, %)	147 (31%)	33 (18%)	114 (40%)	<0.001

^1^ *n/N* (%); Mean (Min–Max); Mean (SD). ^2^ Pearson’s chi-squared test; Wilcoxon ranksum test. * Geometric mean (95% CI).

**Table 2 vaccines-13-01091-t002:** Association of total IgG with malaria infection incidence.

	Malaria Primary Case Definition			Malaria Secondary Case Definition		
Antigen	Crude IRR (95% CI)	IRR Adjusted for Age (95% CI)	*p*-Value for Adjusted IRR	Crude IRR (95% CI)	IRR Adjusted for Age (95% CI)	*p*-Value for Adjusted IRR
AMA1	1.32 (0.92–1.92)	1.36 (0.94–2.00)	0.06	1.27 (0.96–1.68)	1.28 (0.97–1.71)	0.06
CSP	0.84 (0.39–1.79)	0.84 (0.39–1.78)	0.66	0.72 (0.38–1.34)	0.72 (0.38–1.34)	0.31
EBA175 RIII-V	1.21 (0.55–2.73)	1.23 (0.56–2.78)	0.52	0.94 (0.49–1.77)	0.94 (0.49–1.78)	0.84
EBA181 RIII-V	0.67 (0.30–1.46)	0.65 (0.29–1.43)	0.34	0.62 (0.33–1.17)	0.61 (0.32–1.16)	0.17
Etramp4 Ag2	0.50 (0.15–1.52)	0.49 (0.15–1.48)	0.23	0.65 (0.27–1.46)	0.64 (0.27–1.45)	0.32
Etramp5 Ag1	0.57 (0.22–1.35)	0.56 (0.22–1.34)	0.20	0.53 (0.25–1.07)	0.53 (0.25–1.06)	0.08
GEXP18	1.04 (0.50–2.16)	1.03 (0.50–2.14)	0.90	0.88 (0.48–1.59)	0.88 (0.48–1.58)	0.64
GLURP R2	0.69 (0.29–1.61)	0.70 (0.29–1.61)	0.35	0.52 (0.26–1.01)	0.52 (0.25–1.00)	0.04
HSP40 Ag1	0.65 (0.26–1.55)	0.65 (0.26–1.54)	0.38	0.49 (0.23–1.01)	0.49 (0.22–1.00)	0.07
HYP2	0.86 (0.34–2.14)	0.85 (0.34–2.13)	0.76	0.71 (0.32–1.52)	0.70 (0.32–1.51)	0.40
MSP2 CH150	1.10 (0.43–2.93)	1.14 (0.44–3.06)	0.75	0.98 (0.45–2.12)	0.99 (0.45–2.14)	0.98
MSP2 Dd2	1.01 (0.40–2.54)	1.01 (0.41–2.55)	0.96	0.86 (0.41–1.79)	0.86 (0.41–1.78)	0.63
Rh4.2	0.84 (0.37–1.89)	0.85 (0.37–1.90)	0.67	0.68 (0.34–1.31)	0.68 (0.34–1.32)	0.24
Rh5.1	0.61 (0.21–1.57)	0.62 (0.22–1.61)	0.40	0.40 (0.16–0.93)	0.40 (0.16–0.93)	0.07
SBP1	0.58 (0.18–1.72)	0.57 (0.18–1.70)	0.32	0.54 (0.21–1.27)	0.54 (0.21–1.26)	0.17
SEA1	0.43 (0.14–1.21)	0.43 (0.14–1.21)	0.14	0.40 (0.17–0.92)	0.40 (0.16–0.92)	0.05

Malaria incidence rate ratios (IRR) with 95% confidence intervals, corresponding to baseline antibodies, according to different parasitemia thresholds (parasitemia > 0 and > 5000 parasites/µL blood). The results are presented in crude and adjusted for the effect of age.

## Data Availability

The original contributions presented in the study are included in the article, further inquiries can be directed to the corresponding author.

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
