# Peer review of "A Multiplex Serological Assay to Evaluate the Antibody Responses to a Set of Plasmodium falciparum Antigens and Their Protective Role Against Malaria in Children Aged 1.5 to 12 Years Living in a Highly Seasonal Malaria Transmission Area of Burkina Faso"

_vaccines, 2025, doi:10.3390/vaccines13111091_

Round 1
Reviewer 1 Report
Comments and Suggestions for Authors
The immune response in patients with malaria is slow to acquire, species-stage specific,
requires repeated contact to be detected and in the absence of the parasite it disappears.
The study is relevant, the proposed methodology allows achieving the study objectives.
The results provide important information for the diagnosis and control of malaria caused
by Plasmodium falciparum. However, for publication in the journal Vaccine, minor changes must be reviewed. Attached are suggestions for improving the article.

The English is fine and does not require any improvement.
Author Response
Comments 1: In line 41 of the introduction, the statement should be revised because, outside the African continent, not all endemic areas present the circulation of P. falciparum.
Response 1: Agree. corrections have been made.
Comments 2: In the materials and methods section, the text informs the period of sample collection between May and November. How can we ensure that all children included in the study have the same number of malaria episodes and that they were included in the study with a similar incubation period?
Comments 3: Clarify the monitoring period for these children until clinical and parasitological cure is considered.
Response 2 and 3: Thank you for pointing this out. To the comments 2 and 3, clarifications have been made on lines 110 and 190: To ensure comparability, all participants were enrolled at same time point during the peak of the malaria transmission season (September 2020) and followed up prospectively using standardized active and passive surveillance protocols up to April 2021 (malaria low transmission period). Prior to follow-up, all participants were treated and confirmed negative for P. falciparum by PCR. Clinical malaria episode was defined by fever (axillary temperature ≥37.5°C / tympanic temperature ≥38°C or a history of fever within past 24 hours) associated with a positive thick smear/blood film for P. falciparum asexual stages. To avoid double counting, any episode occurring within 28 days of a previous episode was excluded from the incidence calculation, and this 28 days window was also removed from the time-at-risk denominator.
Comments 4- In lines 267 and 268, review the spelling of the words "other-ers" and "combinatiosof"
Comments 5- In table 1, highlight in bold the P values with statistical significance
Comments 6- Improve the resolution of figure 2
Response 4, 5, 6: Thank you for the comments. The corrections have been made to meet the reviewer suggestions. The figure resolution has also been done accordingly.
Reviewer 2 Report
Comments and Suggestions for Authors
The current manuscript employs a combination of cross-sectional and longitudinal strategies to follow children aged 1.5-12 years in a highly seasonal malaria transmission area of Burkina Faso. Microscopy and PCR accuracy were employed for diagnosis in determining infection status. The authors identified that anti-GLURP_R2 and anti-Rh5_1 antibodies are significantly associated with reduced malaria risk, providing potential targets for malaria vaccine development. Besides, machine learning shows certain antibody combination have a good predictive efficacy, offering novel insights for multivalent vaccine design. This study provides valuable data in malaria seroepidemiology, and novel insights into the malaria immune mechanisms and vaccine development. I only have few comments on the methodology and conclusion part.
1)There is a difference in the infection rates between microscopy and PCR (23% vs 31%). The authors need to explain this inconsistency between the two methods.
2)17 antigens were selected, based on previous studies, while the biological significance of each antigen and its specific effect in the Plasmodium life cycle are not mentioned. I recommend providing this information as a supplementary material for a better understanding of the rationale behind the selection process.
3)The conclusion states that "less-studied antibodies showed trends toward reduced risk", but the p-values for some antigens (such as HSP40_ag1) are just marginal. The authors need to clearly distinguish between significance and trend results, to avoid overinterpreting the data.
Author Response
Comments 1: There is a difference in the infection rates between microscopy and PCR (23% vs 31%). The authors need to explain this inconsistency between the two methods.
Response 1: Thank you for pointing this out. The observed difference in infection rates between microscopy and PCR (23% vs 31%) is due to the difference in sensibility between the two diagnostic methods. PCR is molecular technique that is more sensitive and can detect submicroscopic Plasmodium falciparum infection that are not detectable by conventional microscopy. This difference is well documented in the literature, especially in hight-transmission setting, where asymptomatic infections with low parasite density are common.
Comments 2: 17 antigens were selected, based on previous studies, while the biological significance of each antigen and its specific effect in the Plasmodium life cycle are not mentioned. I recommend providing this information as a supplementary material for a better understanding of the rationale behind the selection process.
Response 2: Thank you for the observation. The 16 antigens span multiple stages of the Plasmodium falciparum life cycle, and their choice is based on previous studies demonstrating their immunogenicity or their potential as biomarkers or vaccine targets. To clarify the reader’s understanding, we have added a table in supplementary material (Table S1) describing each antigen, it stages of expression, its known or presumed function, and associated bibliographic references. The salivary antigen gSG6, a protein specific to Anopheles gambiae, was excluded from the analyses in order to restrict the evaluation to antigens exclusively derived from the parasite.
Comments 3: The conclusion states that "less-studied antibodies showed trends toward reduced risk", but the p-values for some antigens (such as HSP40_ag1) are just marginal. The authors need to clearly distinguish between significance and trend results, to avoid over interpreting the data.
Response 3: We recognize the importance of clearly distinguishing statistically significant results from trends. In the revised manuscript, we have clarified in the text that some associations, such as that observed for HSP40_ag1 (p=0.07), represent non-significant trends and should not be interpreted as formal evidence of protection. We have also wording of the conclusion to avoid any over-interpretation, emphasizing the exploratory nature of these results and the need to validate them in other cohorts. Line 321 to 323.
Reviewer 3 Report
Comments and Suggestions for Authors
In this manuscript, the authors evaluated the association between antigen specific IgG responses and protection against P. falciparum infection in children aged 1.5-12 years from Burkina Faso. The manuscript is well structured and easy to read. The experimental part is beyond doubt. The results obtained may find application in the development of test systems and vaccines. The reviewer recommends the manuscript for publication after eliminating a large number of inaccuracies and typos (see below)
- The reviewer did not find the supplementary file (Supplementary Table 1, line 135).
- Lines 59-64: «In addition to these well-studied antigens, less well-studied candidates such as Etramp (Early Transcribed Membrane Proteins), HSP40_Ag1(Heat Shock Protein 40 Antigen 1), HYP2 (Hypothetical Protein 2), GEXP18 (Gametocyte-Exported Protein 18), SEA1 (Sporozoite Exported Antigen 1) and SBP1 (Sporozoite Binding Protein 1) are increasingly being used in more recent approaches, which still require clinical validation to assess their effectiveness as protective antigens in the vulnerable population.»
These statement needs references.
- Line 143: what is the composition of buffer A (please, add)? Why write buffer A if there is no buffer B?
- The caption to Figure 3 needs to be expanded to include sensitivity and specificity.
- In Table 1, it may be necessary to reverse Malaria primary Case definition and Malaria secondary Case definition
Typos:
Lines 38-40: «According to WHO, Burkina Faso was among the 10 high malaria burden countries in 2023, accounted for 3.1% of malaria cases worldwide and 2.7% of malaria deaths in 2023.»
lines 54-55: Reticulocyte Binding Protein Homologue 5
lines 60-61: HSP40.Ag1. the same throughout the manuscript
lines 90-92: «Banfora health District has intense seasonal malaria transmission over the six-month period following seasonal rains from May to November»
line 120: provide the supplier for Giemsa
lines 156-157: Luminex MAGPIX 156 © analyser
Figure 2 on page 8 should be Figure 3 (line 258)
Lines 245-246: «Figure 3 presents the performance of the top 10 antibodies combinations»
Lines 266-268: «Interestingly, P. falciparum infection also AFFECTS (?) antibody profiles, with some antigens elevated in uninfected children, while others like AMA1 were higher in infected children»
Lines 274-276: «Although antibodies against less-investigated antigens such as Etramp4_Ag2 and HSP40_ag1 showed protective trends (IRR<1), these associations did not reach statistical significance and require validation in larger studies.»
Lines 298-300: «higher AMA1 antibody levels were linked to an increased risk of subsequent malaria episodes, as indicated by both crude and age adjusted incidence rate ratio (IRR=1.28-1.36).»
Line 301: rather than
lines 328-329: «Each component contributed distinct protective mechanism»
lines 330-332: «Notably, adding AMA1, a known exposure marker, enhanced the sensitivity of the binary model (GLURP_R2 + Etramp5_ag1) by 14% (sensitivity increase from 0.66 to 0.81).»
line 338: «Studies conducted in low-transmission countries like Gambia….»
line 343: development
line 348: solely
line 364: investigation
Author Response
Comments 1: The reviewer did not find the supplementary file (Supplementary Table 1, line 135).
Response 1: Thank you for the observation, Supplementary Table has been included within associated documents.
Comments 2: Lines 59-64: «In addition to these well-studied antigens, less well-studied candidates such as Etramp (Early Transcribed Membrane Proteins), HSP40_Ag1(Heat Shock Protein 40 Antigen 1), HYP2 (Hypothetical Protein 2), GEXP18 (Gametocyte-Exported Protein 18), SEA1 (Sporozoite Exported Antigen 1) and SBP1 (Sporozoite Binding Protein 1) are increasingly being used in more recent approaches, which still require clinical validation to assess their effectiveness as protective antigens in the vulnerable population.» These statement needs references.
Response 2: References have been inserted in the manuscript Line 73 and Line 75
Comments 3: Line 143: what is the composition of buffer A (please, add)? Why write buffer A if there is no buffer B?
Response 3: The composition of buffer A has been added and additional inputs about buffer B are done in “Antibody Detection Immunoassay” Lines 152 and line 156.
Comments 4: The caption to Figure 3 needs to be expanded to include sensitivity and specificity.
Response 4: Figure 3 is amended accordingly.
Comments 5: In Table 1, it may be necessary to reverse Malaria primary Case definition and Malaria secondary Case definition
Response 5: Amendment is done accordingly.
Comments 6: Typos:
Response 6: Thank you for the observations. corrections have been made.
Lines 38-40: «According to WHO, Burkina Faso was among the 10 high malaria burden countries in 2023, accounted for 3.1% of malaria cases worldwide and 2.7% of malaria deaths in 2023.»
lines 54-55: Reticulocyte Binding Protein Homologue 5
OK
lines 60-61: HSP40.Ag1. the same throughout the manuscript
OK
lines 90-92: «Banfora health District has intense seasonal malaria transmission over the six-month period following seasonal rains from May to November»
OK
line 120: provide the supplier for Giemsa
OK
lines 156-157: Luminex MAGPIX 156 © analyser
OK
Figure 2 on page 8 should be Figure 3 (line 258)
OK
Lines 245-246: «Figure 3 presents the performance of the top 10 antibodies combinations»
OK
Lines 266-268: «Interestingly, P. falciparum infection also AFFECTS (?) antibody profiles, with some antigens elevated in uninfected children, while others like AMA1 were higher in infected children»
OK
Lines 274-276: «Although antibodies against less-investigated antigens such as Etramp4_Ag2 and HSP40_ag1 showed protective trends (IRR<1), these associations did not reach statistical significance and require validation in larger studies.»
OK
Lines 298-300: «higher AMA1 antibody levels were linked to an increased risk of subsequent malaria episodes, as indicated by both crude and age adjusted incidence rate ratio (IRR=1.28-1.36).»
OK
Line 301: rather than
OK
lines 328-329: «Each component contributed distinct protective mechanism»
OK
lines 330-332: «Notably, adding AMA1, a known exposure marker, enhanced the sensitivity of the binary model (GLURP_R2 + Etramp5_ag1) by 14% (sensitivity increase from 0.66 to 0.81).»
OK
line 338: «Studies conducted in low-transmission countries like Gambia….»
OK
line 343: development
OK
line 348: solely
OK
line 364: investigation
OK
Round 2
Reviewer 3 Report
Comments and Suggestions for Authors
The authors took into account the reviewer's suggestions.